# Human farnesyl pyrophosphate synthase is allosterically inhibited by its own product

Jaeok Park[1], Michal Zielinski[1], Alexandr Magder[1], Youla S. Tsantrizos[1,2] & Albert M. Berghuis[1]

Farnesyl pyrophosphate synthase (FPPS) is an enzyme of the mevalonate pathway and a well-established therapeutic target. Recent research has focused around a newly identified druggable pocket near the enzyme's active site. Pharmacological exploitation of this pocket is deemed promising; however, its natural biological function, if any, is yet unknown. Here we report that the product of FPPS, farnesyl pyrophosphate (FPP), can bind to this pocket and lock the enzyme in an inactive state. The $K_d$ for this binding is 5–6 μM, within a catalytically relevant range. These results indicate that FPPS activity is sensitive to the product concentration. Kinetic analysis shows that the enzyme is inhibited through FPP accumulation. Having a specific physiological effector, FPPS is a bona fide allosteric enzyme. This allostery offers an exquisite mechanism for controlling prenyl pyrophosphate levels *in vivo* and thus contributes an additional layer of regulation to the mevalonate pathway.

[1] Department of Biochemistry, McGill University, 3649 Promenade Sir William Osler, Montreal, Quebec, Canada H3G 0B1. [2] Department of Chemistry, McGill University, 801 Sherbrooke Street West, Montreal, Quebec, Canada H3A 0B8. Correspondence and requests for materials should be addressed to A.M.B. (email: albert.berghuis@mcgill.ca).

In mammalian cells, synthesis of many lipids originates from the mevalonate pathway. At the first branching point in this pathway lies farnesyl pyrophosphate synthase (FPPS). FPPS catalyses the sequential condensation of dimethylallyl pyrophosphate (DMAPP) with isopentenyl pyrophosphate (IPP) and the resulting geranyl pyrophosphate (GPP) with another unit of IPP, eventually producing the 15-carbon isoprenoid farnesyl pyrophosphate (FPP; Fig. 1a). FPP serves as a starting substrate for a number of biosynthetic processes. Cholesterol, dolichol and ubiquinone are just a few examples of the numerous downstream products (Fig. 1b). Alternatively, FPP undergoes an additional condensation reaction to produce geranylgeranyl pyrophosphate (GGPP; Fig. 1b). Attachment of a prenyl anchor using FPP or GGPP (viz., prenylation) is essential for proper localization of many proteins. Prenylated proteins constitute up to 2% of the mammalian proteome and are best represented by the small GTPases such as Ras and Rho[1].

The molecular mechanism of FPPS action has been extensively studied[2–4]. An allylic substrate (DMAPP or GPP) binds to the enzyme first, with its pyrophosphate group coordinated between two Asp-rich motifs by three $Mg^{2+}$ ions. The binding of an allylic substrate induces an open-to-closed conformational change in the enzyme, which reshapes its active site cleft and thereby fully forms the IPP-binding site. IPP binding is not metal dependent, occurring mainly through direct interactions between its pyrophosphate head and surrounding protein residues. This binding induces yet another conformational change in the enzyme, which orders the four amino-acid C-terminal tail and seals the active site cavity completely. During catalysis, the prenyl portion of the allylic substrate dissociates as a carbocation and condenses with IPP at its homoallylic double bond. Subsequent proton abstraction by the pyrophosphate leaving group introduces a new carbon double bond in the condensed intermediate, completing the reaction. The proton transfer also facilitates release of the pyrophosphate from the enzyme, which then reverts back to its open state. Translocation of the product (if GPP) to the allylic substrate site or binding of a new DMAPP molecule following its release (if FPP) readies the enzyme for IPP reloading and a subsequent round of catalysis.

As a result of its vast implication for cellular activities, human FPPS has major pharmacological relevance. Inhibition of the enzyme has been well established as the mechanism of action of nitrogen-containing bisphosphonates (N-BPs), blockbuster drugs that are widely used against bone resorption disorders[5]. In addition, there has been growing interest in the anticancer effects of FPPS inhibition. Inhibition of the enzyme deprives cells of FPP and bottlenecks protein prenylation. Without prenylation, oncogenic small GTPases are unable to function and lose their transforming activity[6]. FPPS inhibition also results in accumulation of IPP, which indirectly kills cancer cells by activating $\gamma\delta$ T cells[7]. At present, N-BPs comprise the only class of clinically approved inhibitors of FPPS. As chemically stable substrate analogues, all current N-BP drugs are competitive, active site inhibitors.

Recently, Jahnke et al.[8] identified non-BP FPPS inhibitors that bind to a previously undescribed pocket adjacent to the active site. Despite expanding research efforts for its therapeutic exploitation, the intrinsic function of the newly found druggable pocket has remained elusive. An allosteric regulatory role was proposed, and the biological effector pursued, based on its preference for lipophilic ligands with a negatively charged substituent. However, neither cholesterol and bile acids (downstream metabolites) nor nucleotides and their analogues inhibited the enzyme[8]. More recently, our own efforts identified a different series of non-BP inhibitors targeting the same

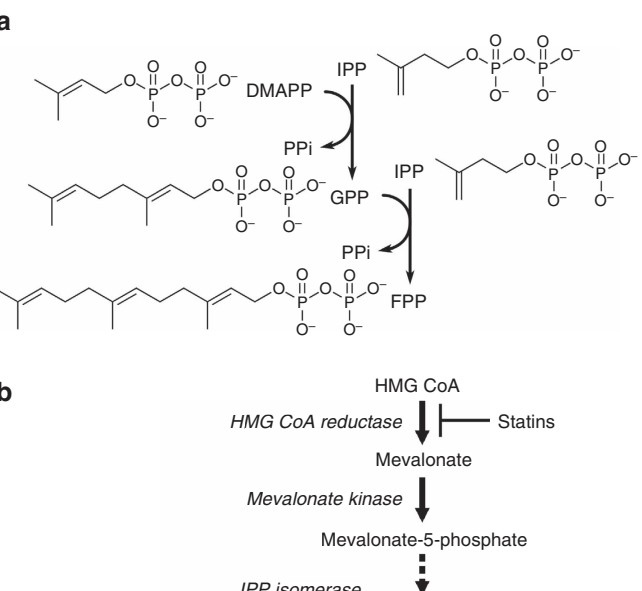

**Figure 1 | FPP synthesis and mevalonate pathway. (a)** Catalytic steps of FPPS reaction. (**b**) Overview of mevalonate pathway and downstream metabolites. Enzymes are shown in Italics. Dotted arrows represent multi-enzyme steps. Sites of intervention by current clinical drugs are indicated. Abbreviations: GGPPS, geranylgeranyl pyrophosphate synthase; HMG CoA, hydroxylmethylglutaryl coenzyme A.

pocket[9,10]. Here we discovered that certain BPs—all of them with bulky lipophilic side chains—could bind to this pocket. This finding raised an interesting possibility: if the identified pocket has a physiological function, the natural allosteric inhibitor might be a prenyl pyrophosphate.

In the present work, we determine a crystal structure of FPPS in complex with FPP. Intriguingly, the product is bound not to the active site, but to the speculated allosteric pocket of the enzyme. Complementary solution studies indicate that this binding occurs in a catalytically relevant concentration range. Indeed, reaction progress kinetic analyses demonstrate product inhibition by FPP. These results strongly suggest that FPP is the physiological allosteric effector of FPPS. The allostery thus provides the enzyme with a negative feedback mechanism, the implication of which extends to the entire mevalonate pathway.

## Results

**Crystal structure of FPPS in complex with FPP.** Human FPPS was crystalized in the presence of FPP, and its X-ray structure was determined at 1.9 Å resolution ($R_{work}/R_{free} = 0.172/0.211$; Table 1). Binding of the product at the previously speculated allosteric pocket was unambiguous based on the electron density and anomalous signals (Fig. 2a). The $\alpha$-phosphate of FPP occupies the entrance of this pocket, engaged in a H-bond with Asn59 and a quadrupole–charge interaction with Phe239 (Fig. 2b). The $\beta$-phosphate sits at the edge of the IPP-binding site, forming salt bridges with Lys57 and Arg60

## Table 1 | Data collection and structure refinement statistics.

|  | Data set 1 (synchrotron) | Data set 2 (home source) |
|---|---|---|
| *Data collection* |  |  |
| Space group | P4₁2₁2 | P4₁2₁2 |
| Cell dimensions |  |  |
| $a, b, c$ (Å) | 110.89, 110.89, 77.48 | 110.70, 110.70, 77.40 |
| $\alpha, \beta, \gamma$ (°) | 90.0, 90.0, 90.0 | 90.0, 90.0, 90.0 |
| Resolution (Å) | 49.59–1.90 (1.95–1.90) | 45.02–2.60 (2.67–2.60) |
| $R_{merge}$ | 0.039 (1.111) | 0.034 (0.192)* |
| $I/\sigma I$ | 31.5 (2.4) | 59.5 (8.5) |
| Redundancy | 9.7 (9.7) | 12.7 (6.0) |
| Completeness (%) | 99.8 (99.0) | 99.1 (96.8) |
|  |  |  |
| *Refinement* |  |  |
| Resolution (Å) | 49.59–1.90 |  |
| No. reflections | 36,427 |  |
| $R_{work}/R_{free}$ | 0.172/0.211 |  |
| No. atoms |  |  |
| Protein | 2,595 |  |
| Ligand/ion | 29 |  |
| Water | 182 |  |
| *B*-factors (Å²) |  |  |
| Protein | 55.5 |  |
| Ligand/ion | 52.6 |  |
| Water | 55.1 |  |
| R.m.s.d.'s |  |  |
| Bond lengths (Å) | 0.029 |  |
| Bond angles (°) | 2.40 |  |

r.m.s.d., root mean squared deviation.
Values in parentheses are for the highest resolution shells.
*If merged.

(Fig. 2b); these residues interact with the α-phosphate of IPP when IPP is bound (Fig. 2c). Additional interactions include those with water molecules and a phosphate ion bound in the IPP site (Fig. 2b). Direct interaction between anionic molecules at this site has been observed multiple times[9–11] and suggests that their charges are neutralized by surrounding protein residues. The tail of FPP extends deep into the allosteric pocket, making tight van der Waals contacts with the protein surface (Fig. 2d). Most of the residues lining this pocket are from helices α_C, α_G, α_H and α_J, which create a long crevice that forms the core of the pocket (Fig. 2a). Tyr10 from α_A covers the open side at the base of this crevice and together with Lys347 shields the FPP tail from bulk solvent (Fig. 2d).

Interestingly, the binding mode of FPP differs from those of the allosteric BPs reported earlier[9,10]. Their pyrophosphate/BP groups interact with the enzyme differently from one another and do not overlap when superimposed (Supplementary Fig. 2). This situation is in marked contrast to the binding of N-BPs at the active site, where the BPs of the inhibitors make identical interactions to those by the pyrophosphate of DMAPP/GPP. Furthermore, FPP binding entails an induced-fit conformational change that has not been observed with other allosteric ligands. The key residues include Tyr10, which swings away from α_C to accommodate the tail end of FPP (Fig. 2e). This change leads to a tilting movement of α_A and allows Lys14 to form new H-bonds with Lys57 and Asn59 (Fig. 2e). The transition also involves Leu62, which rotates toward the FPP tail to provide an additional hydrophobic contact (Fig. 2e). Essentially, the conformational change expands the allosteric pocket and reshapes its surface for better steric complementarity with the long hydrocarbon tail of FPP (Fig. 2f,g). The inherent flexibility in the side chain of Lys347 also contributes to the malleability of this pocket (Fig. 2e). Allosteric binding of FPP was

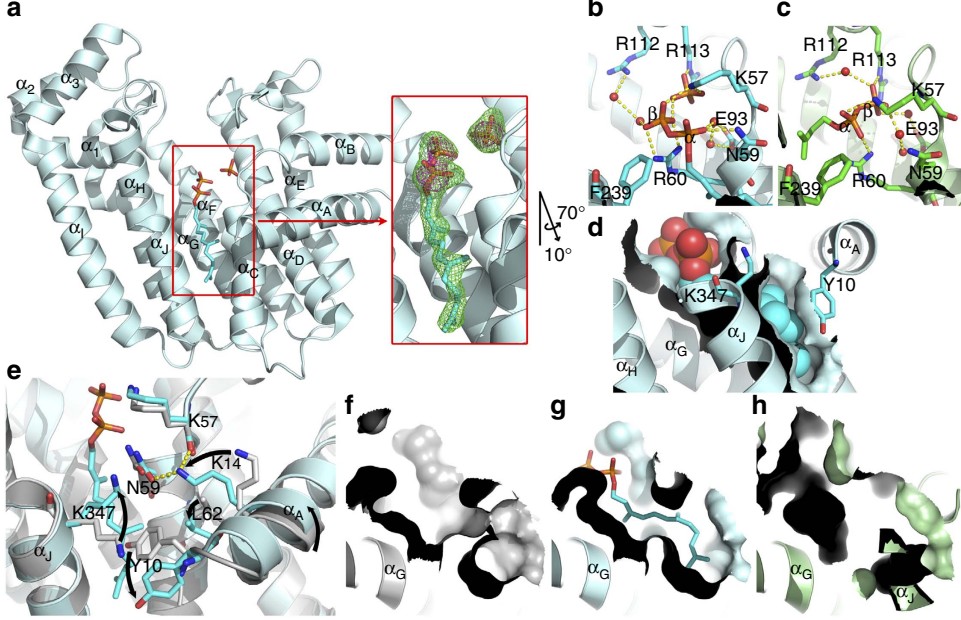

**Figure 2 | Allosteric binding of FPP to FPPS.** (**a**) Overall structure, discovery map (inset, green mesh, $F_o - F_c$ contoured at $3\sigma$), and phosphorus anomalous signal (inset, magenta, contoured at $3\sigma$). Only one subunit (the crystallographic asymmetric unit) is shown for clarity; the biological assembly is a homodimer. A stereo image of the final $2F_o - F_c$ map around the bound ligand is shown in Supplementary Fig. 1. (**b**) Binding interactions by FPP pyrophosphate. (**c**) Binding interactions by IPP pyrophosphate (PDB ID 4H5E). (**d**) FPP in space-filling representation. The surface of the binding pocket is also represented. (**e**) Induced-fit conformational change accompanying FPP binding. The apo-enzyme structure is shown in grey (PDB ID 2F7M). (**f–h**) Allosteric pocket in unliganded, FPP-bound and fully closed states, respectively.

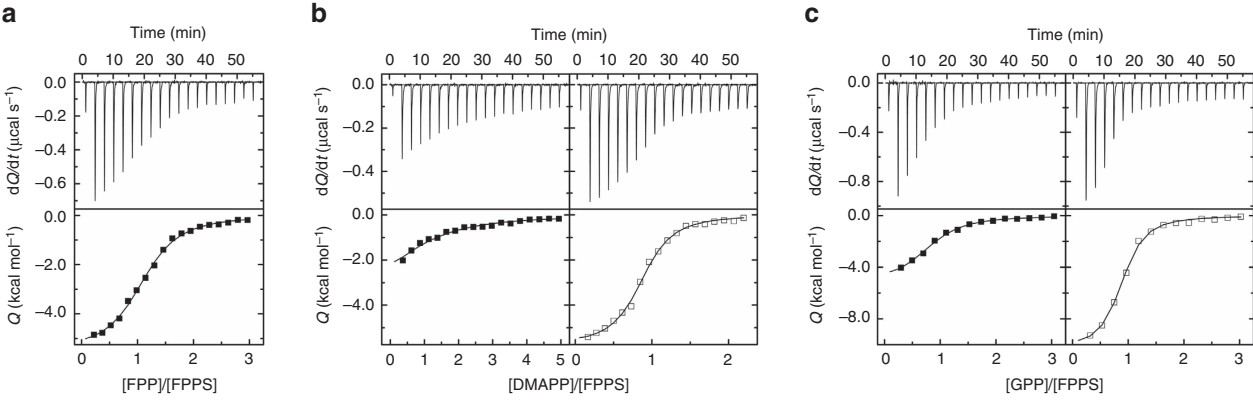

**Figure 3 | Ligand binding to FPPS characterized by ITC.** (**a**) FPP binding in absence of $Mg^{2+}$. The raw thermogram is shown in the upper panel, and the binding isotherm with the fitted curve in the lower panel. (**b**) DMAPP binding in absence (left panels) and presence (right panels) of $Mg^{2+}$. (**c**) GPP binding in absence (left panels) and presence (right panels) of $Mg^{2+}$.

surprising partly because such protein rearrangement could not be predicted from the existing structures.

It has been well established that the conformational transition between the open and closed states of FPPS dictates the progression of its catalytic cycle. As described earlier, the closure of the enzyme enables IPP binding and subsequent catalysis. Opening, on the other hand, facilitates the translocation of GPP or the release of FPP on formation of these products. With FPP bound in the allosteric pocket, the enzyme adopts the open conformational state. Despite the local differences introduced by FPP binding, its overall structure is very similar to that of the apo-enzyme form (Protein Data Bank (PDB) ID 2F7M; Cα root mean squared deviation = 0.21 Å). Of particular significance is that closure of the enzyme brings $\alpha_H/\alpha_J$ closer to $\alpha_C/\alpha_G$ and, as a result, drastically reduces the volume of the allosteric pocket (Fig. 2h). Therefore, allosterically bound FPP can be considered as a molecular wedge that prohibits this conformational transition via steric hindrance. The implication of this insight is clear. FPPS in the unliganded open state is ready to bind a new DMAPP molecule and thus begin another catalytic cycle; but if FPP binds to its allosteric pocket first, the enzyme will stall in its open state and not be able to proceed to the next catalytic step. To assess the physiological relevance of this scenario, it was essential to determine the binding affinity of FPP.

**Thermodynamic characterization of FPP and substrate binding.** The in-solution binding of FPP to FPPS was characterized by isothermal titration calorimetry (ITC). It was shown to be an exothermic process driven by both favourable enthalpy and entropy changes, in which one FPP molecule binds to a single site on the enzyme with a dissociation constant ($K_d$) of 5.3 μM (Fig. 3a; Table 2). The single site deduced here most certainly represents the allosteric pocket based on the present crystal structure. It is conceivable that FPP might also bind to the active site (that is, as if just produced by the enzyme, with its head bound to the IPP site and its tail extended into the allylic substrate site); however, such a binding mode would be energetically unfavourable. We emphasize that this titration experiment was done in the absence of $Mg^{2+}$ or other divalent metal ions, without which FPPS cannot transition into the closed state. With the active site of the enzyme open (and without the pyrophosphate by-product), the tail of FPP would be missing most of its complementary packing surface and exposed to solvent. Interactions with the pyrophosphate moiety would also be suboptimal unlike those seen with the binding of IPP (which occurs with the

enzyme in the closed state). This observation agrees well with the notion that the open conformation of the enzyme facilitates efficient release of the reaction product from the active site.

To compare with the binding affinity of FPP, we next determined those of DMAPP and GPP. It is important to note that while these substrates must bind to the active site (more precisely the allylic substrate site), they should also be able to bind to the allosteric pocket, being structural analogues of FPP that are only shorter in the tail length. We first carried out ITC experiments in the absence of divalent metal ions. Without them, the substrates cannot bind to the allylic substrate site, unable to interact with the negatively charged Asp-rich motifs of the enzyme. The resulting data demonstrated that DMAPP and GPP indeed bind to a single site on the enzyme with $K_d$ values of 43.7 and 7.6 μM, respectively (left panels, Fig. 3b,c; Table 2). The weaker binding compared with that of FPP is due to smaller binding enthalpies ($\Delta H$, Table 2), which likely reflect the decreased hydrophobic effect, as well as fewer van der Waals contacts.

Binding of DMAPP and GPP to FPPS was significantly tighter in the presence of $Mg^{2+}$ (right panels, Fig. 3b,c; Table 2). The $K_d$ values (2.2 and 2.1 μM for DMAPP and GPP, respectively) are in excellent agreement with a previously reported $K_m$ value (2.07 μM for GPP)[3], supporting the interpretation that the substrates were in fact binding to the active site now. Interestingly, the binding affinities of DMAPP and GPP are similar here unlike at the allosteric pocket; the less favourable enthalpy change accompanying the binding of DMAPP is compensated by the more favourable entropic counterpart (Table 2). The one-site-binding pattern observed here is a consequence of the enzyme closure, which renders the allosteric pocket inaccessible to the substrates. Analogous results demonstrating the biased binding (that is, binding exclusively to the allylic substrate site in the presence of $Mg^{2+}$) have been confirmed crystallographically with allosteric BPs[9,12]. In contrast to the binding of DMAPP and GPP, FPP binding was not affected by $Mg^{2+}$ (Table 2). These results suggest that under physiological conditions (present in millimolar concentrations, $Mg^{2+}$ is the second most abundant intracellular ion), DMAPP and GPP bind preferentially to the active site, and FPP to the allosteric site.

**Reaction progress kinetic analysis.** The affinity for the active site binding of DMAPP and GPP to FPPS is less than threefold higher than that for the allosteric binding of FPP (Table 2). The small difference signifies that, if the allosteric binding of

**Table 2 | Thermodynamic parameters determined by ITC.**

| Ligand | $n$ | $K_d$ (µM) | $\Delta H$ (kcal mol$^{-1}$) | $T\Delta S$ (kcal mol$^{-1}$) |
|---|---|---|---|---|
| FPP* | 1.12 ± 0.01 | 5.3 ± 0.4 | − 5.5 ± 0.1 | 1.8 |
| DMAPP* | 1[†] | 43.7 ± 4.3 | − 4.5 ± 0.2 | 1.6 |
| GPP* | 0.80 ± 0.03 | 7.6 ± 0.9 | − 5.3 ± 0.2 | 1.8 |
| DMAPP[‡] | 0.87 ± 0.01 | 2.2 ± 0.2 | − 5.8 ± 0.1 | 2.1 |
| GPP[‡] | 0.79 ± 0.01 | 2.1 ± 0.3 | − 7.7 ± 0.2 | 0.2 |
| FPP[‡] | 1.17 ± 0.02 | 6.0 ± 0.6 | − 5.6 ± 0.1 | 1.7 |

DMAPP, dimethylallyl pyrophosphate; FPP, farnesyl pyrophosphate; GPP, geranyl pyrophosphate.
The experiment was carried out in triplicate.
*Titrated in absence of Mg$^{2+}$.
†The molar binding ratio was not varied during the data fitting process due to a low c value (that is, a weak inflection point).
‡Titrated in presence of Mg$^{2+}$.

FPP indeed inhibits the enzyme, the rate of its reaction would be sensitive to the change in the substrate to product concentration ratio. To probe for time-dependent product inhibition, we analysed FPPS reaction progress, employing the 'same excess' protocol[13]. The evolution of entire reaction was monitored by calorimetry, where two reactions were carried out with different initial concentrations of GPP and IPP (only the second part of the catalytic cycle was examined for simplicity), but with the same difference in the concentrations of the two substrates (that is, [GPP$_0$]–[IPP$_0$] = [excess] = 24 µM; blue and red curves, Fig. 4a). Although GPP and IPP concentrations both decrease as the reaction continues, the change is linked to the reaction stoichiometry (for every molecule of GPP consumed, one molecule of IPP is consumed), and thus [excess] remains constant throughout the full course of the reaction. Therefore, the two reactions represent an identical reaction that started from different time points. At any given point that gives the same amounts of the remaining substrates, there are only two differences between the two reactions: (i) the reaction with the higher initial substrate concentrations has accumulated more FPP; and (ii) the enzyme in this reaction has carried out more turnovers. If the activity of the enzyme had not been affected by these differences, the rate curves for the two reactions would have overlaid onto each other. Instead, the curve for the reaction with the higher initial substrate concentrations traced lower (red, lower panel, Fig. 4a). This result indicates that the enzyme was deactivated over time and/or inhibited by the accumulating product. An additional reaction with an initial amount of FPP produced a depressed rate curve as well (black, Fig. 4a), thus establishing that the reduced catalytic efficiency is due to product inhibition. The enthalpy of reaction ($\Delta H$; equation (1), Methods) was consistent between the three separate reactions at − 22.5, − 22.1 and − 22.3 kcal mol$^{-1}$.

We proceeded to determine some of the kinetic parameters for FPPS reaction. The $K_m$ of IPP was of special interest, since its $K_d$ could not be determined directly by ITC (the catalytically relevant IPP binding is to the FPPS–GPP complex; however, simulating this binding initiates the enzyme reaction). The experiment was carried out with a saturating excess of GPP (~500-fold over enzyme and 10-fold over IPP; blue curve, Fig. 4b) to reduce its analysis to a single-substrate problem. A general steady-state equation that accounts for product inhibition was used (equation (4), Methods). Fitting the data to this model (solid black line, lower panel, Fig. 4b) resulted in a $K_m$ of 1.1 µM, which is comparable to a literature value of 1.8 µM[3]. It is also close to a $K_d$ value (0.9 µM) determined for the binding of IPP to an N-BP-bound FPPS complex[14]. The turnover number ($k_{cat}$) was calculated to be 0.90 s$^{-1}$, slightly higher than previously reported (0.42 s$^{-1}$)[3]. The $K_m$ and $k_{cat}$ values are within the ranges of $10^{-2}$–$10^3$ µM and 0.05–500 s$^{-1}$, respectively, the precise determination of these

parameters, in which the calorimetric instrument used allows for (ref. 15). Interestingly, an analogous experiment carried out in excess of IPP demonstrated significantly reduced enzyme activity (red curves, Fig. 4b). IPP binds also to the allylic substrate site at high concentrations[3], in which case it acts as a competitive inhibitor with respect to DMAPP and GPP. This substrate inhibition would not be relevant physiologically, however, due to the action of IPP isomerase (Fig. 1b).

The new findings of this study update our understanding of the FPPS catalytic cycle; a figure illustrating substrate binding, product release and the conformational transition involved, as well as the measured equilibrium and rate constants, is presented (Fig. 5). Still unknown is the $K_d$ values of the products. Deduced from the crystallographic and thermodynamic data, the $K_d$ of FPP for the active site should be much higher than that for the allosteric pocket. We have attempted to determine the binding affinity of the by-product PPi both in the presence and absence of Mg$^{2+}$; however, the results did not demonstrate an apparent binding event, possibly indicating that the $K_d$ of PPi is also high.

## Discussion

The significance of the mevalonate pathway has been well established. The effectiveness of N-BPs in inducing osteoclast death is a clear testimony to its essentialness. Overactivity of the pathway would also be detrimental as inferred by the many human diseases arising from hyperlipidemic conditions. Naturally, the pathway is kept in check by multiple layers of control mechanisms. It has been long known that the gateway enzyme hydroxylmethylglutaryl coenzyme A reductase (HMGCR; Fig. 1b) is feedback-regulated based on the level of cholesterol both at the transcriptional and post-transcriptional (via enzyme degradation) levels[16]. It was found more recently that the transcription of FPPS is also regulated by the same mechanism used for HMGCR (that is, through the actions of sterol regulatory element binding proteins)[17,18]. Examples of protein level regulation include that of mevalonate kinase (Fig. 1b), which is inhibited by the longer-chain prenyl pyrophosphates GPP, FPP and GGPP[19].

Now, the current study provides data indicating that FPPS is also feedback regulated at the protein level. Significantly, the enzyme is inhibited by its own product and in an allosteric manner. Allostery refers to the phenomenon in which binding of an effector molecule at one site of a protein changes its affinity for a ligand at a spatially distinct second site. FPPS inhibition described in the present report clearly embodies this 'action at a distance' principle. FPP binding at the new druggable site, purely by altering the enzyme's conformational ensemble, interferes with DMAPP binding at the distantly located allylic substrate site (Fig. 5). It is noteworthy that geranylgeranyl

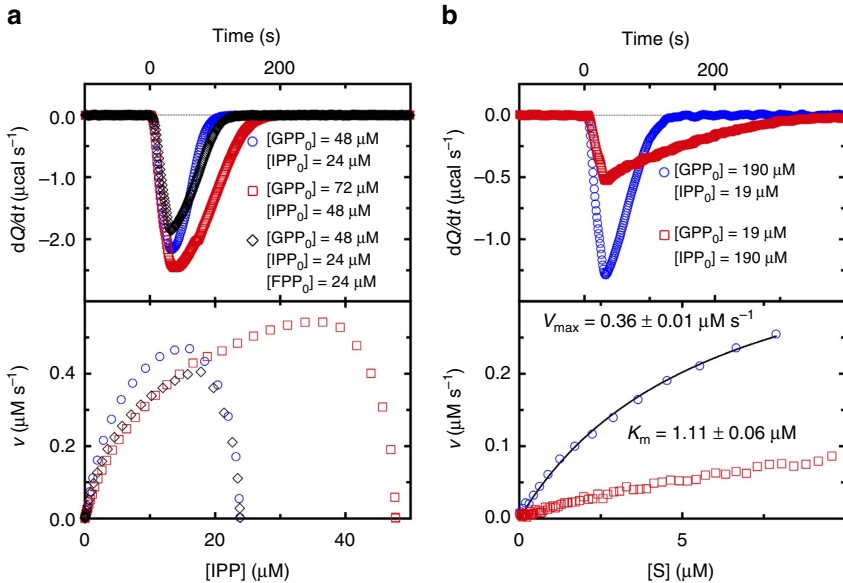

**Figure 4 | Reaction progress kinetic analysis of FPPS.** (**a**) Same excess experiment. Thermograms are shown in the upper panel, and differential rate data generated from the thermograms in the lower panel. The initial substrate and product concentrations are indicated. (**b**) Determination of kinetic parameters. The data from the excess IPP experiment (red) were not regression-analysed due to apparent substrate inhibition.

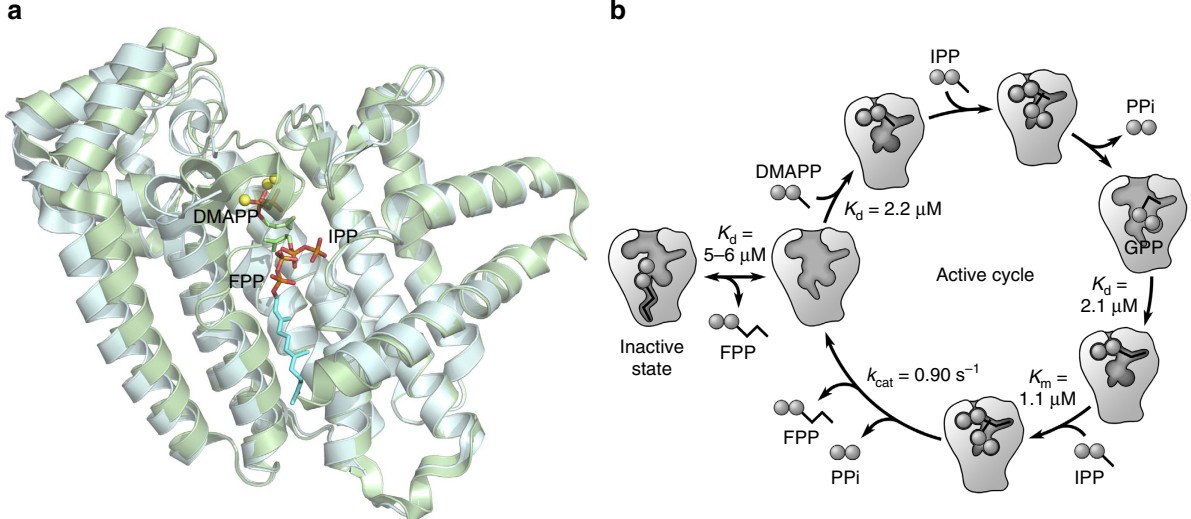

**Figure 5 | Conformational transition and catalytic cycle of FPPS.** (**a**) Superimposition of open (FPP bound, cyan) and closed (substrate bound, green) states. DMAPP was modelled in based on the structures of FPPS in complex with substrate analogues (PDB IDs 1RQI and 4H5E). Yellow spheres are $Mg^{2+}$ ions coordinated to the Asp-rich motifs of the enzyme. (**b**) Schematic representation of FPPS catalytic cycle.

pyrophosphate synthase (GGPPS), the enzyme immediately downstream of FPPS (Fig. 1b), is also inhibited by its own product[20]. This inhibition, however, is not of allosteric nature. GGPP binds in the heart of GGPPS active site and inhibits the enzyme by directly competing with its allylic substrate[20].

Enzymes that are allosterically inhibited by their own products are uncommon. Such an inhibition mechanism allows enzymes to have an immediately responsive feedback process (as opposed to feedback by downstream metabolites) without compromising their catalytic efficiency (active site product inhibition often involves slow product release and/or backward reaction). One of the few examples of allosteric product feedback is found in hexokinase-1 (ref. 21), which catalyses the phosphorylation of glucose by ATP, the first enzymatic step in glucose metabolism. This enzyme is allosterically inhibited by physiological concentrations of its product, glucose-6-phosphate, and thus controls the influx of substrate into the glycolytic pathway. Regulation and modulation of enzyme catalytic activity should be useful in controlling the flux of precursors and products in any metabolic pathway. The cryptic nature of the allosteric pocket in FPPS and the fact that its physiological effector remained unidentified for a long time thus raise an intriguing possibility: allosteric product inhibition might be more prevalent than currently known amongst metabolic enzymes.

Efforts to exploit the allosteric pocket of FPPS as a therapeutic target are actively ongoing. With the pocket dubbed as the 'Achilles' heel' of the enzyme, the enthusiasm in the field is clearly evident. Potent allosteric inhibitors of FPPS may have a wide range of applications, in addition to their potential use as anticancer drugs. For example, they could serve as

cholesterol-lowering agents; nature has already appropriated a similar strategy at the transcriptional level. They may also prove useful against neurodegenerative diseases; a genetic link between elevated levels of FPPS and phosphorylated tau protein, a key factor in neurodegeneration, has been established[9]. Here we revisit the observation that the FPP pyrophosphate bound to the allosteric pocket superimposes poorly with the BPs of our allosteric inhibitors. On the other hand, the first series of allosteric inhibitors discovered by Jahnke et al.[8] is carboxylate based. It is encouraging that the allosteric pocket supports diverse binding poses of different functionalities. Indeed, discovery of new inhibitors based on known drug scaffolds (for example, salicylic acid)[22] and those incorporating distinct functional groups for tissue selectivity (for example, monophosphonate)[23] has been reported very recently. In this light, it is pertinent that FPP binding occurs through an induced-fit mechanism involving expansion of the allosteric pocket. Discovery of additional inhibitors that can exploit such a conformational change is expected.

As an enzyme that catalyses two reactions in a sequential manner, FPPS is a challenging enzyme to study. To make the matter even more complex, its substrates and products are analogues that differ only in their hydrocarbon tail length. Perhaps because of these complications, and despite the bulk of research done, certain aspects of the enzyme have remained undiscovered for decades. In this work, we have demonstrated through a modern kinetic approach that FPPS is inhibited by FPP. Our crystal structure reveals that the product can trap the enzyme in an unreactive state by binding to its allosteric pocket. On the basis of the affinities of the substrates and products measured, this binding should be sensitive to the fluctuating levels of the prenyl pyrophosphates *in vivo*. The allostery thus provides an exquisite means of regulating and fine-tuning these levels. The consequences of our findings on mammalian biology call for future cellular metabolomic studies.

## Methods

**Expression and purification of human FPPS.** A pET-based plasmid encoding human FPPS with an N-terminal His$_6$ tag was transformed into *Escherichia coli* BL21 (DE3) cells. The cells were grown in LB at 37 °C until the OD$_{600}$ of 0.6–0.8 was reached. Expression of the recombinant enzyme was induced by 1 mM isopropylthiogalactoside overnight at 18 °C. To collect the enzyme, the cells were lysed in a buffer containing 50 mM HEPES (pH 7.5), 500 mM NaCl, 2 mM β-mercaptoethanol, 5 mM imidazole and 5% glycerol. The lysate was applied to a metal ion affinity column (Ni-nitrilotriacetic acid), from which the enzyme was eluted with an increasing imidazole gradient. The enzyme containing fractions were pooled and further purified by size-exclusion chromatography (Superdex 200). For storage, the purified enzyme was concentrated to ∼20 mg ml$^{-1}$ by ultrafiltration.

**Isoprenyl pyrophosphates.** DMAPP, IPP, GPP (trans isomer) and FPP (trans,trans isomer) were all purchased from Sigma-Aldrich. When came as a methanol/ammonia solution, the solvent was removed from the sample by desiccating in a centrifugal evaporator. The compounds were dissolved in appropriate buffers for different experiments as described below.

**Crystallization.** FPP was prepared at 5 mM concentration in the final purification buffer (10 mM HEPES (pH 7.5), 500 mM NaCl, 2 mM β-mercaptoethanol and 5% glycerol). MgCl$_2$ was prepared as a 100 mM aqueous solution. FPP and MgCl$_2$ were added to the purified enzyme to give the concentrations of 1 mM FPP, 2 mM MgCl$_2$ and 10 mg ml$^{-1}$ enzyme. A single crystal was obtained at 22 °C by vapour diffusion in a sitting drop composed of 1 μl FPP/MgCl$_2$/enzyme mixture and 1 μl crystallization solution (80 mM TrisHCl (pH 8.5), 1.6 M ammonium phosphate and 20% glycerol).

**Structure determination.** Diffraction data were collected under cryogenic conditions (100 K) first at the home lab with a MicroMax-007 HF generator and a Saturn 944+ charge-coupled device detector, and then at a synchrotron (Beamline 08ID-1, Canadian Light Source, Saskatoon, SK, Canada). The

wavelengths of the X-ray beams used were 1.5418 and 0.97949 Å, respectively. Both data sets were processed with the xia2 package[24]; however, Friedel mates were not merged for the home-source set. Only the synchrotron data were used for structure determination. The initial model was built by a difference Fourier method with a solvent-omitted starting model generated from PDB entry 2F7M. This model was improved through iterative rounds of manual and automated refinement with Coot[25] and REFMAC5 (ref. 26). Ramachandran statistics for the final model show 97% of the residues in the favoured regions and 3% in the allowed regions. An anomalous signal map was calculated from the home-source data with SHELXC[27] and ANODE[28]. The phase information used in this calculation was obtained from the structure model refined against the synchrotron data. Data collection and structure refinement statistics are summarized in Table 1.

**Binding assay.** Binding experiments were carried out at 30 °C with a MicroCal iTC200 system. The purified enzyme was dialyzed overnight against the binding assay buffer (50 mM HEPES (pH 7.5), 150 mM NaCl, 2 mM β-mercaptoethanol and 5% glycerol). Ligand (DMAPP, GPP and FPP) and MgCl$_2$ solutions were prepared in the used dialysate. Each titration experiment consisted of a first 1 μl injection followed by 18 2 μl injections of a ligand solution into the 204.1 μl calorimetric cell loaded with the enzyme solution. The concentration of the enzyme in the cell was 100 μM (in monomers), and those of the ligands in the titration syringe ranged from 1 to 2 mM. When added, the concentration of MgCl$_2$ was 5 mM. Heats of dilution were measured by injecting the ligands into the buffer alone and subtracted from the corresponding titration data. The data were fitted to the single-site-binding model implemented in the Origin software package provided with the ITC instrument.

**Reaction assay.** Reaction calorimetry was also carried out at 30 °C with a MicroCal iTC200 system. The enzyme and substrates were prepared in the reaction buffer (50 mM HEPES (pH 7.5), 150 mM NaCl, 2 mM MgCl$_2$, 2 mM β-mercaptoethanol and 5% glycerol) in the same way described for the binding experiments. The reactions were assayed by a single injection method[15], where 10 μl substrate solution of both GPP and IPP was injected into the calorimetric cell containing the enzyme. When added, FPP was preincubated together with the enzyme. The concentration of the enzyme in the cell was 400–500 nM, and those of the substrates in the syringe were 0.4–4 mM. Heats of dilution were measured and subtracted from the actual reaction data. The raw data were processed with the Enzyme Assay module of the Origin package, and plots of reaction rate as a function of substrate concentration were generated (five data points were binned for each data point displayed). Briefly, The molar reaction enthalpy (ΔH) was determined first based on the relationship:

$$Q = n\,\Delta H = [S_0]\,V\,\Delta H, \tag{1}$$

in which $Q$ is the total heat associated with producing $n$ moles of product, $[S_0]$ is the starting concentration of the limiting substrate and $V$ is the volume of the calorimetric cell. $Q$ was calculated by integrating the thermal power (d$Q$/d$t$) measured over the complete course of reaction, whereas $[S_0]$ and $V$ were known. Once $\Delta H$ was determined, the substrate concentration ($[S]$) could be determined as a function of time as described in the equation:

$$[S] = [S_0] - \frac{\int_0^t \frac{dQ}{dt}\,dt}{V\,\Delta H}. \tag{2}$$

The rate of reaction could be calculated from d$Q$/d$t$ for any given time point:

$$\text{Rate} = \frac{d[P]}{dt} = -\frac{d[S]}{dt} = \frac{1}{V\,\Delta H}\frac{dQ}{dt}. \tag{3}$$

For kinetic measurements, the temporal experimental heat flow must be monitored accurately. To ensure that the heat flow measured was not convoluted with the rate of heat transfer through the reactor wall, the experimental data were mathematically corrected by the apply time constant function implemented in the Origin software. $K_m$ and $V_{max}$ values were determined by fitting the rate data to the following equation[29]:

$$\frac{d[P]}{dt} = \frac{\frac{V_{max}\,[S]}{(1 - K_m/K_p)}}{\frac{[S_0] + K_p}{(K_p/K_m - 1)} + [S]}. \tag{4}$$

The value of 6 μM was substituted for the product affinity term ($K_p$) as determined in the binding assay. To carry out analysis with only the portion of the reaction exhibiting steady-state behaviour, data obtained during the induction period were omitted.

**Data availability.** Sequence information on human FPPS is available in the UniProt Knowledgebase under accession code P14324. The PDB accession codes 1RQI, 2F7M, 4H5E, 4LPG and 4QXS were used in this study. Coordinates and structure factor of the structure reported here have been deposited into the Protein Data Bank under accession code 5JA0. All other relevant data are available from the corresponding author upon reasonable request.

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

## Acknowledgements

We thank the beamline personnel at the Canadian Light Source for data collection. This work was supported by grants from the Canadian Institute of Health Research to Y.S.T. (CIHR-126062) and A.M.B. (MOP-114889), and the Fonds de recherche du Québec—Nature et technologies to both Y.S.T. and A.M.B. (FRQ-NT PR-181227). A.M.B. holds a Canada Research Chair in Structural Biology.

## Author contributions

J.P. designed the study and performed all experiments unless noted otherwise. M.Z. and A.M. participated as undergraduate project students: M.Z. set-up the crystallization trays, and A.M. carried out part of the ITC experiments. J.P. analysed all experimental data and wrote the manuscript together with Y.S.T. and A.M.B.

## Additional information

**Competing financial interests:** The authors declare no competing financial interests.

