## [Peer review file · Nature Communications]

REVIEWERS' COMMENTS:

Reviewer #1 (Remarks to the Author):

This is a nice paper in which the authors find that FPPS is inhibited by its product, FPP. Given the importance of FPPS as a drug target, this new discovery will be of interest to many groups.

Reviewer #2 (Remarks to the Author):

This is a definitive study that identifies FPP as an allosteric inhibitor of the ubiquitous metabolic enzyme FPP synthase. Previous studies have identified the presence of an allosteric binding site, distinct from the catalytic site, but the natural ligand was not known.

The studies are accurately described and the results are clearly presented and interpreted. As the authors point out, this is a particularly challenging enzyme to study kinetically. However, the calculated kinetic parameters are consistent with other, previously published kinetic measurements and the molecular interactions determined from the crystal structures are highly informative. Minor points on the accuracy of the Introduction: 1) Line 67 onwards - the major pharmacological relevance of FPPS inhibition is the potent effect of bisphosphonates, a blockbuster class of drugs. 2) Line 72 - IPP does not directly kill cancer cells but can be converted to Apppl, an inhibitor of mitochondrial function. Whether this occurs in cancer cells in vivo is controversial, therefore the direct anti-cancer actions of these drugs on vivo is still highly questionable. Methods: presumably the all-trans isomer of FPP was used, but this should be stated.

AUTHORS' RESPONSE TO THE REVIEW COMMENTS

Reviewer #1 (Remarks to the Author):

This is a nice paper in which the authors find that FPPS is inhibited by its product, FPP. Given the importance of FPPS as a drug target, this new discovery will be of interest to many groups.

Response:

We are grateful for the time and efforts by the reviewer in reviewing our manuscript, as well as the favourable review itself. We hope that our discovery will indeed be of interest to many other research groups.

Reviewer #2 (Remarks to the Author):

This is a definitive study that identifies FPP as an allosteric inhibitor of the ubiquitous metabolic enzyme FPP synthase. Previous studies have identified the presence of an allosteric binding site, distinct from the catalytic site, but the natural ligand was not known.

The studies are accurately described and the results are clearly presented and interpreted. As the authors point out, this is a particularly challenging enzyme to study kinetically. However, the calculated kinetic parameters are consistent with other, previously published kinetic measurements and the molecular interactions determined from the crystal structures are highly informative. Minor points on the accuracy of the Introduction: 1) Line 67 onwards - the major pharmacological relevance of FPPS inhibition is the potent effect of bisphosphonates, a blockbuster class of drugs. 2) Line 72 - IPP does not directly kill cancer cells but can be converted to ApppI, an inhibitor of mitochondrial function. Whether this occurs in cancer cells in vivo is controversial, therefore the direct anti-cancer actions of these drugs on vivo is still highly questionable. Methods: presumably the all-trans isomer of FPP was used, but this should be stated.

Response:

We greatly appreciate the reviewer's understanding of the topic and insightful comments. We have revised the manuscript to address all three specific points raised by the reviewer.

1) Regarding the major pharmacological relevance of FPPS inhibition:

We have rewritten the 3rd paragraph of the introduction to clearly indicate that the major pharmacological relevance of FPPS inhibition is the potent antiresorptive effects of bisphosphonates. In comparison, the anticancer effects of FPPS inhibition is described as "growing" interest in the same paragraph.

2) Regarding the cancer cell killing effects of IPP accumulation:

We have removed the statement that IPP kills cancer cells by inducing apoptosis since this mechanism is controversial in vivo as the reviewer pointed out. Activation of gamma delta T cells by IPP accumulation is described as an indirect mechanism.

3) Regarding the use of FPP isomers:

We have indeed used trans isomers of both GPP and FPP in our experiments. This information is now clearly stated in the methods section.